# New Fuzzing Biases for Action Policy Testing

**Primary Keywords:** *None*

## Abstract

Testing was recently proposed as a method to gain trust in learned action policies in classical planning. Test cases in this setting are states generated by a fuzzing process that performs random walks from the initial state. A fuzzing bias attempts to bias these random walks towards policy bugs, that is, states where the policy performs sub-optimally. Prior work explored a simple fuzzing bias based on policy-trace cost. Here, we investigate this topic more deeply. We introduce three new fuzzing biases based on analyses of policy-trace *shape*, estimating whether a trace is close to looping back on itself, whether it contains detours, and whether its goal-distance surface does not smoothly decline. Our experiments with two kinds of neural action policies show that these new biases improve bug-finding capabilities in many cases.

## Introduction

Learned action policies, in particular ones represented by neural networks, are gaining traction in AI (e.g., Mnih et al. 2013; Silver et al. 2016, 2018), and are being intensively explored in AI planning (Issakkimuthu, Fern, and Tadepalli 2018; Groshev et al. 2018; Garg, Bajpai, and Mausam 2019; Toyer et al. 2020; Karia and Srivastava 2021; Ståhlberg, Bonet, and Geffner 2022a,b). But how to gain trust that such a policy will yield desirable (or at least non-fatal) behavior?

Policy *testing* is one natural answer to this question. Most work so far views the environment as an adversary, controlled by the testing mechanism in a way that tries to make the system fail (e.g., Dreossi et al. 2015; Akazaki et al. 2018; Koren et al. 2018; Ernst et al. 2019; Lee et al. 2020). While this can be quite useful, it does not distinguish whether or not the failure is actually due to bad policy decisions, or is unavoidable given the environment behavior.

Recent work (Steinmetz et al. 2022) addresses this in the context of classical planning, defining test cases as states $s$, and policy *bugs* as states on which the policy is sub-optimal (a better policy exists for $s$, e.g., avoiding failure). Given a test case $s$, test *oracles* are used to detect whether $s$ is a bug, by evaluating sufficient criteria that avoid the need for a full optimal planning process (Eisenhut et al. 2023). To generate test cases in the first place, a *fuzzing* process iteratively builds up a *pool* of test states by conducting random walks from the initial state. A *fuzzing bias* attempts to bias these random walks towards policy bugs, thus making the overall testing machinery more effective in finding bugs. Specifically, we consider *bias functions* mapping states to numbers, with higher numbers indicating more promising states.

Prior work (Steinmetz et al. 2022) only explored a simple bias function, mapping a state $s$ to the summed-up action cost of the policy trace in $s$. Here, we investigate this topic more deeply. We introduce three new fuzzing biases that aim at analyzing the policy-trace *shape*. In what we call the **loopiness bias**, we measure how close a trace is to looping back on itself, a typical reason for failure to reach the goal (in dead-end free domains, this is indeed the only reason). In the **detour bias**, we measure the largest detour (the most sub-optimal segment) along the trace. In the **surface bias**, we consider goal-distance along the trace as its surface, and measure the degree to which that surface does not smoothly decline along the path. All three measures rely on estimates of the distance between states (on the trace, respectively to the goal). We include an analysis of the idealized setting using the exact distance $h^\star$; in practice, we use the delete relaxation heuristic $h^{\mathrm{FF}}$ (Hoffmann and Nebel 2001).

Using $h^{\mathrm{FF}}$ instead of $h^\star$ of course makes the measurements inaccurate. In particular, "detours" and "goal distance increases" will often simply be due to $h^{\mathrm{FF}}$ estimation errors, but biases may still contain information useful for bug-finding. Indeed, this is what our experiments show. We consider two kinds of neural action policies: ASNets (Toyer et al. 2018, 2020), and Ståhlberg et al.'s (2022a; 2022b) GNN policies. We test these policies across a set of domains where they perform well. Our new fuzzing biases improve bug-finding capabilities in many of these cases.

## Background

An **FDR planning task** is a tuple $\Pi = \langle \mathcal{V}, \mathcal{A}, I, G \rangle$. $\mathcal{V}$ is a finite set of **variables**, each variable $V \in \mathcal{V}$ has a finite **domain** $\mathrm{dom}(V)$. A **partial state** $p$ is a variable assignment over some variables $\mathcal{V}(p) \subseteq \mathcal{V}$; $p[V]$ is the value assigned to $V \in \mathcal{V}(p)$ in $p$. Given $U \subseteq \mathcal{V}$, $p[U]$ denotes $p$ restricted to $U$. A partial state $s$ is a **state** if $\mathcal{V}(s) = \mathcal{V}$; $\mathcal{S}$ denotes the set of all states. $I$ is an **initial state**. $G$ is a partial state called **goal**, and a state $s$ is a **goal state** if $G = s[\mathcal{V}(G)]$.

$\mathcal{A}$ is a finite set of **actions**, $a \in \mathcal{A}$ is defined by its **precondition** $\mathrm{pre}(a)$, **effect** $\mathrm{eff}(a)$, and **cost** $\mathrm{cost}(a) \in \mathbb{R}_0^+$. Preconditions and effects are partial states. $a \in \mathcal{A}$ is **applicable** in $s \in \mathcal{S}$ if $\mathrm{pre}(a) = s[\mathcal{V}(\mathrm{pre}(a))]$. Given $s \in \mathcal{S}$,

$\mathcal{A}[s]$ is the set of all actions applicable in $s$. The **resulting state** of applying an applicable action $a$ in a state $s$ is $s' = a[\![s]\!]$ s.t. $s'[V] = \text{eff}(a)[V]$ for every $V \in \mathcal{V}(\text{eff}(a))$, and $s'[V] = s[V]$ otherwise. A sequence of actions $\phi = \langle a_1, \ldots, a_n \rangle$ is **applicable** in $s_0 \in \mathcal{S}$ if there are states $s_1, \ldots, s_n$ s.t. $a_i$ is applicable in $s_{i-1}$ and $s_i = a_i[\![s_{i-1}]\!]$, the resulting state is $\phi[\![s_0]\!] = s_n$. Given $\phi$ applicable in $s$, the sequence $\langle s_0, s_1, \ldots, s_n \rangle$ of the aforementioned states is called **intermediate state sequence** of $\phi$. The cost of $\phi$ is defined as $\text{cost}(\phi) = \sum_{i=1}^{n} \text{cost}(a_i)$. $\phi$ applicable in $s$ s.t. $G = \phi[\![s]\!][\mathcal{V}(G)]$ is called $s$-**plan**, and $I$-plan is called **plan**. We consider **heuristic functions** $h$ as estimators for costs of optimal $s$-plans, denoted as $h(s)$, and optimal paths between two states $s$ and $t$, denoted $h(s, t)$. We assume all heuristics are safe, and $h^\star(s)$ and $h^\star(s, t)$ denote optimal costs.

Deterministic **policies** $\pi \colon \mathcal{S} \to \mathcal{A} \cup \{\emptyset\}$ map states to applicable actions or null ($\emptyset$) if there is no applicable action. The **run** (or **trace**) $\sigma^\pi(s)$ of a policy $\pi$ on state $s$, is an action sequence $\sigma^\pi(s) = \langle a_1, \ldots, a_n \rangle$ applicable in $s$ (with its intermediate state sequence $\langle s_0, \ldots, s_n \rangle$) s.t. $a_i = \pi(s_{i-1})$ for all $1 \leq i \leq n$, all intermediate states differ from each other, and no $s_i$ is a goal state unless $i = n$. The **cost** of $\sigma^\pi(s)$ is $\text{cost}^\pi(s) = \text{cost}(\sigma^\pi(s))$ if $\sigma^\pi(s)[\![s]\!]$ is a goal state, and $\text{cost}^\pi(s) = \infty$ otherwise; $\text{cost}_{s_0}^\pi(s_i, s_j)\,(i < j)$ denotes the cost of the subsequence of $\sigma^\pi(s_0)$ between $s_i$ and $s_j$.

We adopt the testing framework of Steinmetz et al. (2022): Given a task $\Pi$ and a policy $\pi$, a state $t$ is called a **bug** in $\pi$ if $\text{cost}^\pi(t) > h^\star(t)$.[1] To find bugs, policy testing (1) generates a pool $\mathcal{P} \subseteq \mathcal{S}$ of test states using a fuzzing process, and (2) runs test oracles on each pool state. Here, we focus on (1) only. For (2), we fix the state-of-the-art oracle of Eisenhut et al. (2023) ("BMO-100 + Aras/EHC").

The fuzzing process proposed by Steinmetz et al. uses random walks from $I$ guided by a bias $\mathcal{B}$, which is a function mapping states to numbers where higher numbers indicate more promising states (more likely to lead to a bug state). The fuzzer evaluates $\mathcal{B}$ on every successor state $s$, and obtains a probability distribution over the respective actions by normalizing the biases $\mathcal{B}(s)$. We adopt this fuzzing process here, though with some refinements as described in the next section. Steinmetz et al. only explored a simple bias $\mathcal{B}^\pi(s) = \text{cost}^\pi(s)$: the cost of the policy run. The fuzzing process with $\mathcal{B}^\pi$ prefers to explore states with more costly policy traces. Under the assumption that such states are more likely to be bugs, this makes sense, and indeed it often increases the number of detected bugs in comparison to a uniform random fuzzer. Nevertheless, this bias encapsulates a rather naïve notion of "good" vs. "bad" policy traces. Here, we investigate this topic more deeply, through new fuzzing biases that aim at analyzing the shape of policy runs.

## The Fuzzing Process

We design a refined version of the fuzzer introduced by Steinmetz et al. (2022), see Algorithm 1: Within a runtime limit $T$, we iteratively generate a pool $\mathcal{P}$ of maximal size $N$ by random walks of maximal length $L$. Random walks

---

[1]It covers both "quantitative" ($\pi$ is sub-optimal on $t$) and "qualitative" ($\pi$ does not solve solvable $t$) bugs used by Steinmetz et al.

---

**Algorithm 1:** Refined version of Steinmetz et al.'s fuzzer.

**Input:** Action policy $\pi$, bias function $\mathcal{B}$, initial state $I$
**Parameters:** Max. pool size $N$, runtime limit $T$, max. walk length $L$, budgets $B_{\text{expansion}}$ and $B_{\text{state}}$
**Output:** Pool $\mathcal{P}$ of test states

1   $\mathcal{P} \leftarrow \{I\}$;
2   **while** $|\mathcal{P}| < N \wedge \text{runtime} \leq T$ **do**
3     $l \leftarrow \text{uniformRandom}(\{1, \ldots, L\})$;
4     $s_0 \leftarrow \text{uniformRandom}(\mathcal{P})$; $s \leftarrow s_0$;
5     **for** $i = 1 \ldots l$ **do**
6       $S' \leftarrow \emptyset$; $j \leftarrow B_{\text{expansion}}$;
7       **for** $s' \in \{s[\![a]\!] \mid a \in A[s], G \neq s[\![a]\!][\mathcal{V}(G)]\}$ **do**
8         **if** $h^{\max}(s') = \infty$ **then continue**;
9         **if** $\mathcal{B}$ requires computing $\text{cost}_{s'}^\pi$ **then**
10           **if** $j \leq 0$ **then break**;
11           $\sigma \leftarrow \text{partialRun}(\pi, s', \min(j, B_{\text{state}}))$;
12           $b \leftarrow (\infty \text{ if } \pi \text{ fails in } \sigma \text{ else } \mathcal{B}(s'))$;
13           $j \leftarrow j - |\sigma|$;
14         **else** $b \leftarrow \mathcal{B}(s')$;
15         **if** $b > -\infty$ **then** $S' \leftarrow S' \cup \{(s', b)\}$;
16       **if** $S' = \emptyset$ **then goto** 2;
17       $s \leftarrow \text{weightedSelect}(S')$;
18     $\mathcal{P} \leftarrow \mathcal{P} \cup \{s\}$;

---

start from a randomly selected pool state $s_0$ and are steered by a bias function $\mathcal{B}$, influencing which successor state $s'$ is selected in each step of the random walk. Steinmetz et al. only considered the bias $\mathcal{B}^\pi$ returning the cost of the policy run. Here, we use bias functions $\mathcal{B} \colon \mathcal{S} \to \mathbb{R} \cup \{-\infty, \infty\}$ allowing negative bias values as well as $-\infty$ and $\infty$.

The application of $\mathcal{B}$ is still straightforward. In each step of the random walk, we compute a set $S'$ containing pairs $(s', b)$ of successor states $s'$ and bias values $b$, ignoring goals and states with bias $-\infty$. We then pick one of the states in $S'$ as follows (function weightedSelect): if there is any $(s', \infty)$ in $S'$, we pick one $s'$ among all such $s'$ uniformly at random. Otherwise, we determine the minimal bias value $b_{\min}$. If $b_{\min} < 0$, we increase all $b$ by $|b_{\min}|$ so that all $b$ are nonnegative. Finally, we normalize the bias values to sum up to 1 and select $s'$ according to this probability distribution.

For biases that require computing the cost of partial policy runs ($\mathcal{B}^\pi$ and all biases introduced next), our fuzzer includes further additions. For each successor state $s'$, we use the function $\text{partialRun}(s', \pi, k)$ to obtain the partial policy trace $\sigma$ of $\pi$ on $s'$ of at most $k$ steps, which is then analyzed by $\mathcal{B}$. If we can infer from $\sigma$ that $\pi$ fails on $s'$, we ignore $\mathcal{B}(s')$ and assign $b = \infty$ instead, i.e., we consider only policy runs that could lead to a goal state. As we ignore goal states, this also means $\mathcal{B}(s')$ is not computed if $\sigma$ is empty.

Since evaluating policies is often very time-consuming, we introduce simple mechanisms bounding the number of policy evaluations. We limit the number of steps of individual policy runs by $B_{\text{state}}$, and we limit the overall number of policy steps per successor state by $B_{\text{expansion}}$.

## Bias Functions Measuring Policy-Trace Shape

The only bias from prior work preferred policy traces with high cost, which may be due to bad policy behavior, but may

just as well simply be due to high plan cost. Here, we instead introduce measures of policy-trace *shape*, encapsulating deeper notions of policy-trace quality. Specifically, we introduce what we call the **loopiness bias**, **detour bias**, and **surface bias**. All three biases consider pairs of states $s_i$ and $s_j$ on the policy trace, and maximize a measure of "trace-shape badness" over all such state pairs. These measures all rely on heuristic functions $h$ estimating state distances; we consider both, the idealized setting where $h = h^\star$, as well as $h = h^{\text{FF}}$ which we use in practice. As part of our experiments, we run the idealized biases with $h^\star$ where feasible, to evaluate the impact of using $h^{\text{FF}}$ instead.

We next introduce each bias in turn. In the following, let $s$ denote the current state reached by a random walk as per Algorithm 1, let $\langle a_1, \ldots, a_n \rangle$ denote a non-empty partial run of the policy $\pi$ on $s$, i.e., the first $n$ steps of the policy run $\sigma^\pi(s)$, and let $\langle s_0, \ldots, s_n \rangle$ denote the respective intermediate state sequence where $s = s_0$. We omit all proofs here; they are included in the accompanying technical report.

**Loopiness Bias** The loopiness bias is geared at estimating whether, at some point along the policy trace, that trace is close to forming a loop. The bias is defined as follows:

$$\mathcal{B}_h^{\text{loop}}(s) = \max_{0 \le i < j \le n} \text{cost}_s^\pi(s_i, s_j) - h(s_j, s_i).$$

For every state $s_i$ and later state $s_j$ along the policy trace, we measure (a) the cost of the policy path from $s_i$ to $s_j$, minus (b) the estimated cost of a plan from $s_j$ back to $s_i$. This measure will be high if $s_i$ and $s_j$ are distant on the policy trace, but only few steps are needed to go back from $s_j$ to $s_i$. Note here that, while one may be tempted to use only (b) and minimize over its values, (a) is useful because states close to each other on the path may naturally have very small values for (b). Indeed, in *invertible* planning tasks where every action has a direct inverse of the same cost, this is necessarily the case as (b) will be the cost of a single action for every state pair $s_i$ and $s_j$ with $j = i + 1$ along the trace.

Invertible tasks also form a special case regarding the outcome of the maximization in the idealized setting:

**Proposition 1.** *In invertible planning tasks,* $\mathcal{B}_{h^\star}^{\text{loop}}(s) = \text{cost}_s^\pi(s_0, s_n) - h^\star(s_n, s_0).$

Of course, this does not hold if we replace $h^\star$ with $h^{\text{FF}}$. We considered to save computational effort (avoiding maximization) for $\mathcal{B}_{h^\star}^{\text{loop}}$ by detecting invertible planning tasks automatically (e.g., Hoffmann 2005), but these automatic tests failed to find any invertible tasks in our benchmark set.

**Detour Bias** The detour bias aims to directly measure whether a policy trace is sub-optimal. We consider every segment from $s_i$ to $s_j$ on the trace and compare its cost under the policy to the estimated cost of a path from $s_i$ to $s_j$:

$$\mathcal{B}_h^{\text{detour}}(s) = \max_{0 \le i < j \le n} \text{cost}_s^\pi(s_i, s_j) - h(s_i, s_j)$$

Large values of $\text{cost}_s^\pi(s_i, s_j) - h(s_i, s_j)$ indicate that the policy trace "takes a detour" from $s_i$ to $s_j$. Note that, syntactically, the only difference to the loopiness bias is the order of $s_i$ and $s_j$ in the call to $h$. That difference can lead to

arbitrary differences in bias value however (even for $h^\star$ and in invertible planning tasks, as there may be a much cheaper path from $s_j$ to $s_i$ than the inverted policy path).

For $h^\star$ (but not for $h^{\text{FF}}$), maximization here is not needed as any subpath detour is contained in the overall path:

**Proposition 2.** $\mathcal{B}_{h^\star}^{\text{detour}}(s) = \text{cost}_s^\pi(s_0, s_n) - h^\star(s_0, s_n).$

It is furthermore easy to see that the idealized detour bias carries exact information about bugs and their optimality gap (note that unsolvable states can by definition not be bugs):

**Proposition 3.** *Let $s$ be a solvable state. If $\mathcal{B}_{h^\star}^{\text{detour}}(s) > 0$, then $s$ is a bug. If $s$ is a bug and $\langle a_1, \ldots, a_n \rangle$ reaches the goal, then $\mathcal{B}_{h^\star}^{\text{detour}}(s) > 0$ and there exists a plan for $s$ whose cost is at most $\text{cost}_s^\pi(s_0, s_n) - \mathcal{B}_{h^\star}^{\text{detour}}(s)$.*

**Surface Bias** Search space surface has previously been investigated as a way of examining the quality of heuristic functions (e.g., Hoffmann 2005, 2011). Here we take a related perspective, considering the surface of the policy trace when taking $h$-values as the vertical dimension. On an optimal path to the goal and if $h = h^\star$, this surface will decline smoothly, i.e., it will reduce by $\text{cost}(a_i)$ in each step. The surface bias aims at measuring the degree to which this is not the case: $\mathcal{B}_h^{\text{surf}}(s) =$

$$\begin{cases} -\infty & h(s) = \infty \\ \max_{0 \le i < j \le n} \text{cost}_s^\pi(s_i, s_j) - (h(s_i) - h(s_j)) & h(s) \ne \infty \end{cases}$$

If $h(s) = \infty$, $s$ is unsolvable, so we return the minimal bias value $-\infty$. Otherwise, this measure includes (a) the cost of the policy path from $s_i$ to $s_j$, minus (b) the decrease in $h$ value from $s_i$ to $s_j$. If (a) is larger than (b), then the policy-trace surface between $s_i$ and $s_j$ does not decline as smoothly as it should. In particular, if $s_j$ is far from $s_i$ on the path, but is estimated to be further from the goal, i.e., $h(s_j) > h(s_i)$, the surface bias will be very high, indicating that the trace is meandering around, not making progress towards the goal. Of course, outside the idealized $h^\star$ setting such an indication may be misguided merely due to local minima in $h$. Again, maximization here is never needed for $h = h^\star$:

**Proposition 4.** $\mathcal{B}_{h^\star}^{\text{surf}}(s) = \text{cost}_s^\pi(s_0, s_n) - (h^\star(s_0) - h^\star(s_n)).$

The bug information in the idealized setting is even stronger than for the detour bias:

**Proposition 5.** *Let $s$ be a solvable state. If $\mathcal{B}_{h^\star}^{\text{surf}}(s) > 0$, then $s$ is a bug. If $s$ is a bug and $\langle a_1, \ldots, a_n \rangle$ reaches the goal, then $\mathcal{B}_{h^\star}^{\text{surf}}(s) > 0$ and the optimal plan cost for $s$ is exactly $\text{cost}_s^\pi(s_0, s_n) - \mathcal{B}_{h^\star}^{\text{surf}}(s)$.*

The surface bias also has a desirable property in a non-idealized setting:

**Proposition 6.** *For any safe heuristic function $h$, $\mathcal{B}_h^{\text{surf}}(s) = \infty$ only if $\langle a_1, \ldots, a_n \rangle$ ends in a dead-end state.*

## Experiments

For our experiments, we extend the testing framework by Eisenhut et al. (2023), which builds on the one by Steinmetz et al. (2022). As in their work, we test ASNets policies (Toyer et al. 2018, 2020), but we use our own, less resource

| Policy | Domain | #Π | $S$ | Oracle coverage (practical biases) | | | | | $S$ | Oracle coverage (all biases) | | | | | | | |
|---|---|---|---|---|---|---|---|---|---|---|---|---|---|---|---|---|---|
| | | | | $\mathcal{B}^0$ | $\mathcal{B}^\pi$ | $\mathcal{B}^{\text{detour}}_{h^{\text{FF}}}$ | $\mathcal{B}^{\text{surf}}_{h^{\text{FF}}}$ | $\mathcal{B}^{\text{loop}}_{h^{\text{FF}}}$ | | $\mathcal{B}^0$ | $\mathcal{B}^\pi$ | $\mathcal{B}^{\text{detour}}_{h^{\text{FF}}}$ | $\mathcal{B}^{\text{detour}}_{h^\star}$ | $\mathcal{B}^{\text{surf}}_{h^{\text{FF}}}$ | $\mathcal{B}^{\text{surf}}_{h^\star}$ | $\mathcal{B}^{\text{loop}}_{h^{\text{FF}}}$ | $\mathcal{B}^{\text{loop}}_{h^\star}$ |
| ASNets | Blocks | 21 | 1488.0 | **7.6** | 7.2 | 7.3 | 7.4 | **7.6** | 21.0 | 4.8 | 4.8 | 4.8 | 4.8 | 4.8 | 4.8 | 4.8 | 4.8 |
| | Elevator | 58 | 5279.8 | 75.0 | 89.8 | **92.6** | 90.6 | 91.4 | 341.2 | 45.4 | 66.3 | **77.1** | 73.6 | 68.4 | 68.4 | 71.0 | 73.6 |
| | Floortile | 14 | 1400.0 | **11.3** | 5.2 | 5.2 | 4.0 | 4.1 | 182.4 | 8.5 | 4.0 | 4.0 | 3.9 | 3.9 | **31.4** | 3.9 | 3.9 |
| | Gripper | 35 | 2967.6 | **0.0** | **0.0** | **0.0** | **0.0** | **0.0** | 752.4 | 0.0 | 0.0 | 0.0 | 0.0 | 0.0 | 0.0 | 0.0 | 0.0 |
| | MBlocks | 16 | 1477.0 | **15.1** | 13.9 | 14.0 | 14.3 | 13.5 | 524.6 | 13.3 | 12.5 | 13.3 | 13.5 | 13.5 | **66.4** | 12.9 | 13.5 |
| | Satellite | 15 | 1210.0 | 67.1 | 83.8 | 86.0 | **89.1** | 85.9 | 388.8 | 48.4 | 69.7 | 74.3 | 81.8 | 78.3 | 68.3 | 73.7 | **83.1** |
| | Scanalyzer | 48 | 2796.2 | 42.4 | 68.2 | **73.5** | 70.1 | 71.1 | 543.0 | 26.3 | 57.9 | 62.8 | **89.3** | 59.0 | 56.1 | 59.1 | 59.1 |
| | Storage | 19 | 1137.4 | 47.6 | 78.7 | 79.5 | 80.0 | **81.4** | 486.2 | 32.0 | 59.3 | 58.8 | 59.2 | **61.3** | 59.6 | **61.3** | 59.2 |
| | Transport | 26 | 1946.8 | 62.1 | 79.5 | 81.9 | 82.6 | **83.0** | 944.4 | 36.5 | 66.3 | 71.1 | 70.5 | 71.4 | 63.2 | **73.5** | 70.5 |
| | VisitAll | 30 | 1812.0 | 67.6 | 81.1 | **85.9** | 80.7 | 81.6 | 566.4 | 44.2 | 56.1 | 64.1 | **64.4** | 56.4 | 53.7 | 58.4 | 58.5 |
| | Woodwork | 38 | 3700.6 | 55.6 | 78.2 | 81.1 | **88.5** | 75.9 | 1779.4 | 46.7 | 68.4 | 71.6 | 68.8 | **83.6** | 66.3 | 66.0 | 64.7 |
| GNNs | Blocks | 35 | 2944.6 | 34.9 | 34.6 | 36.0 | 36.1 | **36.4** | 1646.6 | 14.1 | 14.7 | 15.4 | **28.2** | 15.7 | 15.0 | 16.0 | **28.2** |
| | Delivery | 26 | 2113.8 | 18.2 | 17.5 | 21.6 | 17.3 | **23.3** | 1925.2 | 15.3 | 13.5 | 16.9 | 22.2 | 14.6 | 14.6 | 20.7 | **22.5** |
| | Gripper | 17 | 986.8 | 24.1 | **25.7** | 21.4 | 21.1 | 24.4 | 312.0 | 26.7 | 27.9 | 20.5 | 18.8 | 9.9 | 7.5 | **28.3** | 22.5 |
| | Miconic | 67 | 4020.0 | 34.2 | 32.3 | **38.1** | 34.5 | 32.6 | 3886.4 | 33.1 | 31.4 | 37.0 | **38.5** | 33.3 | 30.8 | 31.9 | 29.6 |
| | Reward | 20 | 1329.2 | 39.6 | 38.7 | 41.3 | **45.8** | 41.5 | 1298.4 | 39.7 | 38.7 | 40.4 | 43.8 | **45.8** | 42.8 | 41.8 | 39.0 |
| | Visitall | 16 | 431.0 | 41.3 | **59.4** | 51.9 | 51.1 | 58.6 | 206.2 | 17.7 | **33.6** | 19.6 | 32.2 | 18.3 | 20.5 | 28.9 | 25.8 |

Table 1: Oracle coverage (in %) for two sets of biases. #Π is the number of problem instances per domain; we only include tasks where $\pi$ solves the initial state. $S$ is the overall number of considered states (sum over domain). Oracle coverage is the percentage of states in the respective combined pool (of size $S$) identified as bugs (see text). Results averaged over five runs.

intensive, C implementation of ASNets and a more comprehensive benchmark set. We also consider GNN policies (Ståhlberg, Bonet, and Geffner 2022a,b). Here, we test the original policies provided by Ståhlberg, Bonet, and Geffner (2022b) on a subset of the original benchmark set. Both kinds of policies are run in separate processes with their own memory limit of 2 GiB for ASNets and 8 GiB for GNNs. The testing engine uses a limit of 2 GiB. A cluster with Intel E5-2660 processors was used. If this paper is accepted, we will make code and benchmarks publicly available.

We compare our new biases (in $h^{\text{FF}}$ and $h^\star$ configuration) with two baselines: the constant bias $\mathcal{B}^0 = 0$ and $\mathcal{B}^\pi$ defined as $\mathcal{B}^\pi(s) = \text{cost}^\pi_s(s_0, s_n)$. Given our additions to the fuzzer, $\mathcal{B}^\pi$ differs from Steinmetz et al.'s definition in that it only computes the cost of *partial* runs.

For each bias $\mathcal{B}$ and planning task, we run the fuzzer from Algorithm 1 for at most one hour to generate a pool $\mathcal{P}$ of up to $N = 100$ states. We limit the length of random walks to $L = 5$ and use the budgets $B_{\text{state}} = 50$ and $B_{\text{expansion}} = 200$. For the test oracle, as mentioned before we use the best-performing oracle of Eisenhut et al. (2023). We run this oracle for at most two hours per test pool $\mathcal{P}$, to determine how many states in $\mathcal{P}$ can be classified as bugs.

We refer to the percentage of tested pool states (across the entire domain) classified as bugs as *oracle coverage*. To foster comparison of oracle coverage across biases, we use the same number of test states across all biases for each problem instance. If we have, e.g., 90 pool states for $\mathcal{B}_1$ but only 50 for $\mathcal{B}_2$, we consider only the first 50 pool states for $\mathcal{B}_1$.

Table 1 shows the results. The left part shows the comparison of practical biases (not using $h^\star$). In general, the behavior is highly domain specific and there is no bias that works best universally. However, in 7 of the 11 domains considered for ASNets and in 4 out of 6 domains for GNNs, one of the new biases achieves a higher oracle coverage than both baselines. The new biases perform significantly better than $\mathcal{B}^0$ in most cases, but usually provide only a slight advantage over $\mathcal{B}^\pi$. Note that the oracle coverage in general varies greatly between domains. For example, the coverage for ASNets in Elevator is not below 75% across all biases, but we do not detect a single bug state for ASNets in Gripper (we assume the policy is optimal). We are able to increase the coverage by 10 percentage points (from $\mathcal{B}^\pi$) in Woodworking, but the difference between all biases in Blocks is negligible.

The right part shows the results when including all biases. As this includes $h^\star$ based biases, the number of considered states $S$ is mostly much lower (e.g., the pools for ASNets in Blocksworld contain the initial state only). While there are cases where the $h^\star$ variant outperforms its $h^{\text{FF}}$ proxy (e.g. $\mathcal{B}^{\text{detour}}_{h^\star}$ for ASNets in Scanalyzer), this is not the case in general which indicates that $h^{\text{FF}}$ works reasonably well here in replacing $h^\star$. Note also that, here, there is only a single case (GNNs in Visitall) where a baseline achieves better results than all configurations of our new biases.

## Conclusion

With the proliferation of learned action policies, how to meaningfully test them is becoming very relevant. We contribute more advanced fuzzing biases for policy testing in classical planning, measuring path-shape features instead of merely path cost, with promising results.

We do not believe that much more can be gained by different fuzzing biases. The primary challenges for policy testing in our view remain fault analysis (which specific policy decisions cause sub-optimality), extension to richer planning paradigms, and connections to policy re-training.

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
