# OpenReview forum: "New Fuzzing Biases for Action Policy Testing"
_icaps-conference.org/ICAPS/2024/Conference — ICAPS 2024_

### Official Review · Reviewer_2L6r · 2024-01-20

**Significance And Importance:** 2
**Soundness:** 4
**Novelty:** 3
**Clarity:** 4
**Overall Evaluation:** 2
**Confidence:** 3

**Weaknesses:**

2: No major or minor weaknesses.

**Contributions Of The Paper:**

The paper addresses the topic of testing a learned action policy via fuzzing. Here the goal is to find states where a suboptimal action is taken. The approach collects a set of states via random walks guided by a bias function and then tests these states with test oracle for suboptimality. Previous work had only used very basic bias functions. This paper introduces three new more sophisticated bias functions and shows that each of them is beneficial for at least one benchmark domain.

**Ethical Considerations:**

(1) Not Applicable: The paper does not have any ethical considerations to address

**Nomination For Best Paper:**

No

**Questions For Authors:**

1) Why don't you treat quantitative and qualitative suboptimality separately? In my opinion, there is a big difference between a policy that fails to reach the goal and a policy that reaches the goal but requires suboptimal cost.
2) Have you considered or even tested other aggregation functions apart from "max"?
3) Roughly how large was the standard deviation for the results in Table 1?
4) Do your results for the GNN domains match the results in their original paper? I think a short discussion would benefit the paper.
5) How much faster or memory-efficient is your C implementation of ASNets compared to the original Python implementation? Were you able to reproduce the results from the original paper with your implementation?

**Reproducibility:**

4: Authors promise to release code and domains (whichever apply).

**Strengths Of The Paper:**

The paper addresses a relevant problem. It is well written and the experiments are well designed. The theory is motivated and explained well and all contributions look sound to me.

**Weaknesses Of The Paper:**

The experimental results are a bit underwhelming. However, this could also be due to the experiment design since it's unclear how much any method could improve over the previous best methods. When computing oracle coverage, couldn't you ignore the states proven optimal by some method? It would be good to see whether a domain even allows for further improvement, or not.

The introduction briefly mentions test oracles, but to understand the relevance of the work, it would be good to include at least a superficial discussion of test oracles and why no optimal planning processes is needed for proving a state suboptimal.

It's a pity that there's no space for the proofs, not even proof sketches. To make the paper self-contained, I recommend buying an extra page and including proof sketches.


Minor comments:

Define "safety" of a heuristic.

The use of h^max in Algorithm 1, line 8 should be motivated in the text. And a citation for h^max would be good.

Bias paragraphs: since they span multiple paragraphs, they should be subsections instead.

---

> ### Author Rebuttal · Authors · 2024-01-26
>
> Thank you for the detailed review.
>
> You are right in that it is not clear how much a method can improve over other methods, e.g., for ASNets and Gripper, the policy seems to be optimal on all reachable states, so that no bias could achieve a higher coverage than 0%. However, ignoring states on which the policy performs optimally when computing coverage does not solve the problem here. We measure the quality of biases based on oracle coverage, so if a bias chooses more states on which the policy performs optimally than other biases, the quality of the bias is lower. So, if we ignored all states on which the policy performs optimally, we would effectively only test how well the oracle confirms bug states to be bugs, but not how good the biases are.
>
> We will buy an extra page if the conference gives that option.
>
> Regarding your questions:
> 1) We agree that this distinction makes sense in principle. For our purpose of evaluating biases, making the distinction is not necessary though, and doing so would require a much more complicated results presentation and discussion, going substantially beyond the page limit.
> 2) We have not conducted experiments with other aggregation functions. We considered other possibilities, such as using "sum", however one possible advantage of using "max" instead could be that it is less dependent on the sheer length of the (partial) policy trace we analyze.
> 3) We assume you mean the standard deviation σ with respect to the 5 repetitions of the experiment. The average σ is about 2.7% with respect to the left part and about 4.2% with respect to the right part of Table 1 (when averaging the σ values for all entries). There are significant differences with respect to domains and policies.
> 4) While we added some glue code to integrate GNN policies into our testing framework, we use their code and test the policies trained by the original authors (Stahlberg et al., 2022).
> 5) With respect to policy evaluation, our C implementation is up to one order of magnitude faster and about 4 times more memory efficient. The quality of the resulting policies depends on the training, but we were able to train policies that solve roughly the same number of planning tasks.

---

### Official Review · Reviewer_NPMn · 2024-01-22

**Significance And Importance:** 2
**Soundness:** 3
**Novelty:** 2
**Clarity:** 3
**Overall Evaluation:** 2
**Confidence:** 3

**Weaknesses:**

1: Minor weaknesses that are easily fixable.

**Contributions Of The Paper:**

Thanks to the authors for their responses to our reviews.  Their explanations and answers make sense, and I remain convinced that the paper should be accepted.   While acknowledging that the paper is space-constrained, it would still be nice to see a bit more discussion of next steps (such as provided in the author response) added to the very short second paragraph in the paper's conclusion.

Original review:

This paper reports on further results regarding heuristic biases applied to fuzzing, as a means to evaluate "neural action policies," meaning neural nets trained to implement the mapping from state to a next action.  The fuzzing process generates a set of candidate initial states as test cases, to which the trained NN is then applied.
The most significant result in the paper is largely a negative result:  fancier fuzzing heuristics of three different types, based on properties of the policy trace starting in a given state, are shown to result in only limited improvements in testing coverage.

**Ethical Considerations:**

(1) Not Applicable: The paper does not have any ethical considerations to address

**Nomination For Best Paper:**

No

**Questions For Authors:**

The main question I have regarding this paper is how general the result is.  In particular, what is the basis for the assertion that not much more can be gained by considering alternative fuzzing biases?  Are the variants evaluated in this paper somehow representative of a space of alternatives?  Are the authors rethinking the utility of fuzzing biases at all?  Or is it that addressing the other challenges cited in the conclusion is more likely to lead to improved testing performance?

**Reproducibility:**

2: Some details are missing, but the paper still appears to be replicable with some effort.

**Strengths Of The Paper:**

The paper is clearly written.  While it is somewhat notation heavy, definitions are clear and well-motivated.  The reported result is of some scientific interest, in providing evidence of diminishing returns for increasingly informed fuzzing heuristics.   Relevant work is cited.  The evaluation is sufficient.

**Weaknesses Of The Paper:**

There are some places definitions would help, at least for non-specialists.  What is "FDR planning?"  A brief explanation of test oracles would help, too.

---

> ### Author Rebuttal · Authors · 2024-01-26
>
> Thanks for your review.
>
> If accepted and if it is possible to buy an extra page, we will add further explanations (including for test oracles) as you suggested.
>
> Regarding your questions:
> Based on our experiments, we cannot exclude the possibility that there are other biases that work significantly better in practice than the ones we tried so far, especially if one considers other domains and other policies. However, one aspect that supports our assumption is that we experimented with biases (the h* variants) that have presumably almost ideal theoretical properties, namely that they carry exact information about whether a state is a bug and even on the "optimality gap" (see Propositions 3 and 5). Therefore, if we keep the set-up we have settled on now and only consider partial policy traces, it should be hard to do better. We do not rethink the utility of fuzzing biases at all; the comparison with B^0 shows that biases can be useful. So, the main message of the conclusion is indeed rather that we believe that addressing the further challenges we outlined is likely a more fruitful investment of our time. However, the aim here is not only to improve testing performance, but also to pursue further steps, e.g, we think the current framework is robust enough to support targeted retraining.

---

### Official Review · Reviewer_P8kB · 2024-01-22

**Significance And Importance:** 2
**Soundness:** 3
**Novelty:** 2
**Clarity:** 3
**Overall Evaluation:** 2
**Confidence:** 2

**Weaknesses:**

1: Minor weaknesses that are easily fixable.

**Contributions Of The Paper:**

This paper focuses on the problem of testing the optimality of general action policies for classical planning problems. In particular, the objective is to find planning states on which the policy solves the problem with a sub-optimal sequence of actions. These states are also called bugs. As done in previous works, a random walk guided by a bias function is used to find these states. The paper contributes by introducing three new biases: the loopiness bias, the detour bias, and the surface bias.
An experimental analysis shows that the new biases are overall more effective at discovering bugs than the baselines.

**Ethical Considerations:**

(1) Not Applicable: The paper does not have any ethical considerations to address

**Nomination For Best Paper:**

No

**Questions For Authors:**

1) Is there any particular reason why you choose h^FF? Can you spend some words on the difference between using an admissible vs a non-admissible heuristic for your biases?

2) Can you briefly explain how the oracle you used works?

3) You state (line 224) “That difference can lead to arbitrary differences in bias value however (even for h* and in invertible planning tasks, as there may be a much cheaper path from s_j to s_i than the inverted policy path)”. I thought that for invertible tasks and h* the term h(s_i, s_j) is equal to h(s_j, s_i), making B^detour equal to B^loop (in this case). Can you clarify this?

**Reproducibility:**

4: Authors promise to release code and domains (whichever apply).

**Strengths Of The Paper:**

This paper tackles an interesting problem. The technique proposed would be very helpful to re-train policies.

The adopted approach is clear and easy to follow.

The paper is sound as far as I could check, and the experimental analysis seems solid.

**Weaknesses Of The Paper:**

The clarity of the background section could be improved. Here are some suggestions:

- Use lowercase letters for single variables, e.g., “V” at line 75 and onwards. Keep uppercase letters for sets, such as “U”.
- Differentiate the “p[]” notation for single variables and sets of variables. You use p[v] to denote a value and p[U] (U set of variables) to denote a partial state. Use a different notation to avoid confusion.
- At line 85, why write pre(a) = s[V(pre(a)]? If pre(a) is a partial state and s a complete state, then you can write pre(a) \subseteq s. The same applies to goal G at line 96 and onwards.
- Is it necessary to use \phi to indicate a sequence of actions? This notation is only used in one paragraph. Also, consider changing the letter “\phi” as it commonly denotes logic formulas.

I suggest spending a few words explaining how the oracle works.

When explaining Algorithm 1, I suggest adding references to the lines of code in the text.
I suggest changing the notation “h^max” as it commonly denotes the admissible h max heuristic.

Consider moving “BMO-100 + Aras/EHC” to the experiments section.

---

> ### Author Rebuttal · Authors · 2024-01-26
>
> Thank you for the constructive review.
>
> We will implement your write-up suggestions to further improve clarity.
> We, indeed, mean the standard (admissible) h^max heuristic in line 8 of Algorithm 1. It is used as a simple test to avoid visiting dead end states, which per our definition cannot be bugs. We will add a brief explanation in the respective paragraph.
>
> Regarding your questions:
> 1) We consider FF as the first obvious choice here as it is based on extracting a relaxed plan which often leads to good estimates (although not admissible). However, we also tried the LMCut heuristic and observed very similar behavior, but FF is faster and allows to test more states. Moreover, our current data does not suggest that choosing a more sophisticated heuristic would lead to better
> biases.
>
> 2) The oracle we use is a so-called bound-maintenance oracle (BMO). BMOs maintain upper bounds u(t) >= h*(t) for test states t and aim at reducing these upper bounds by comparing the states using state dominance functions. Clearly, if cost^pi(t) > u(t) >= h*(t), t is a bug. The initial source of upper bounds u(t) are policy runs and search. This is combined with external tools, e.g., it also uses the plan improvement tool Aras to search for a better plan in case a state cannot be shown to be a bug otherwise. Provided that it is possible to buy an extra page, we will also include a brief description of the oracle in the paper.
>
> 3) Yes, in the special case of invertible tasks and h*, B^detour is equal to B^loop. If either the task is not invertible or we are using h neq h*, then the biases are different. We will reformulate accordingly. Thanks for pointing this out.

---

### Meta-Review · Area_Chair_jdKu · 2024-02-02

**Recommendation:** Accept (Oral)
**Confidence:** 5

**Metareview:**

The paper aims at finding states where a suboptimal action is taken by a learned policy. The authors use fuzzing to collect states to be tested via random walks guided by a bias function and introduce three novel bias functions.

All the reviews agree that the paper is well written and tackles an interesting problem. Despite some doubts on the significance of the experimental results (that show modest improvements with respect to simpler bias functions), all the reviewers are in favor of acceptance.

**Ethical Considerations:**

(1) Not Applicable: The paper does not have any ethical considerations to address